# Vitamin D/Bone Mineral Density and Triglyceride Paradoxes Seen in African Americans: A Cross-Sectional Study and Review of the Literature

**DOI:** 10.3390/ijms25021305

**Published:** 2024-01-21

**Authors:** Christopher M. Stevens, Sushil K. Jain

**Affiliations:** Department of Pediatrics, Louisiana State University Health Sciences Center, 1501 Kings Highway, Shreveport, LA 71103, USA; christopher.stevens@lsuhs.edu

**Keywords:** 25(OH)VD (25-hydroxyvitamin D), VDBP (vitamin D binding protein), VE (vitamin E), TG (triglycerides), BMD (bone mineral density), AA (African Americans), PTH (parathyroid hormone), LPL (lipoprotein lipase), APO (apolipoprotein), HDL (high density lipoprotein), H_2_S (hydrogen sulfide), CBS (cystathionine β-synthase), CSE (cystathionine-γ-lyase)

## Abstract

Vitamin D is known to have a positive effect on bone health. Despite the greater frequency of vitamin D deficiency in African Americans (AA), they have a higher bone mineral density (BMD) compared to whites, demonstrating a disconnect between BMD and vitamin D levels in AA. Another intriguing relationship seen in AA is the triglyceride (TG) paradox, an unusual phenomenon in which a normal TG status is observed even when patients house conditions known to be characterized by high TG levels, such as Type II diabetes. To the best of our knowledge, no study has examined whether these two paradoxical relationships exist simultaneously in AA subjects with Type II diabetes. In this study, we compared levels of blood markers, including HbA1c, TG, and vitamin D, measured as serum 25-hydroxyvitamin D [25(OH)VD] µM/mL, [25(OH)VD]/TG, calcium, and BMD in AA (*n* = 56) and white (*n* = 26) subjects with Type II diabetes to see whether these relationships exist concurrently. We found that AA subjects had significantly lower TG and [25(OH)VD] levels and a significantly higher BMD status compared to white subjects, even when the ages, BMI, duration of diabetes, HbA1c, and calcium levels were similar between the two groups. This demonstrates that these two paradoxical relationships exist simultaneously in Type II diabetic AA subjects. In addition to these findings, we discuss the current hypotheses in the literature that attempt to explain why these two intriguing relationships exist. This review also discusses four novel hypotheses, such as altered circulating levels and the potential role of estrogen and hydrogen sulfide on BMD and HMG-CoA reductase as a possible contributor to the TG paradox in AA subjects. This manuscript demonstrates that there are still many unanswered questions regarding these two paradoxical relationships and further research is needed to determine why they exist and how they can be implemented to improve healthcare.

## 1. Introduction

A paradox is defined as a statement, proposition, or situation that seems self-contradictory but upon further evaluation is found to possess some truth [1]. Vitamin D is a micronutrient essential for bone health as it promotes calcium and phosphorus absorption, the main components of hydroxyapatite, and its quantitation is widely used to determine the risk for osteoporosis. Epidemiologic studies consistently show that African Americans (AA) have a higher prevalence of vitamin D deficiency compared to whites [2,3,4,5,6,7]. However, AA are known to have higher bone mineral density (BMD) compared to other races, demonstrating a disconnect between bone health and vitamin D levels in AA [2,5,6,8,9]. Therefore, vitamin D levels and bone health in AA present a paradox. In addition, another unexplainable relationship seen in AA is the triglyceride (TG) paradox, an unusual phenomenon in which AA consistently express a normal TG status even when they have conditions known to be characterized by high TG levels, such as Type II diabetes, insulin resistance, and cardiovascular disease [10,11,12,13]. In this study, we compared levels of blood markers, including HbA1c, TG, and vitamin D, measured as serum 25-hydroxyvitamin D [25(OH)VD] uM/mL, [25(OH)VD]/TG, calcium, and BMD in AA and white subjects with Type II diabetes to see whether these relationships exist concurrently. To the best of our knowledge, no study has examined whether these two paradoxical relationships exist simultaneously in AA subjects with Type II diabetes, a condition known to be associated with decreased vitamin D and BMD and increased TG levels [14,15,16].

Additionally, a review of the current hypotheses in the literature that attempt to explain why these two intriguing relationships exist in AA is presented. We also discuss four novel hypotheses regarding why these paradoxes might be present. Our first hypothesis discusses whether there is a potential link between the vitamin D and TG paradoxes, and that quantifying vitamin D as a ratio of serum vitamin D to total serum TG results in a better correlation with BMD compared to using circulating levels of [25(OH)VD] per milliliter of serum to assess vitamin D status. Next, we explain how the increased levels of estrogen, a hormone that promotes bone maturation, in AA may explain why AA have increased BMD compared to other races despite having diminished vitamin D status. Our third hypothesis discusses the possible connection between hydrogen sulfide (H_2_S), vitamin D, and adiponectin levels and how this may lead to an increased BMD status in AA. Lastly, we mention how AA may have a certain HMG-CoA reductase gene variant that results in decreased TG levels, thus leading to a possible explanation for the TG paradox.

## 2. Methods

The prior approval of the Louisiana State University (LSU) Institution Review Board for Human Experimentation (IRB) was obtained on 11 January 2016 (protocol number: H-09-073) before the start of subject enrollment in the study. A written consent form and HIPPA form, approved by the IRB, were collected from each Type II diabetic patient aged 30–55 years. These subjects were asked to participate in this study while attending a regular diabetes clinic for routine checkups. Subjects who participated in a prior clinical trial were asked if they wanted to participate in the current study. AA and whites of both sexes were invited to participate in this study. Exclusion criteria included if patients had a history of cardiovascular disease, sickle cell disease, insulin treatment, or metabolic disorder, including hypothyroidism and hyperthyroidism. Subjects with a history of cardiovascular disease, sickle cell disease, insulin treatment, and metabolic disorders were excluded because these disorders are known to impact lipid metabolism and may falsely influence the outcome of the results. Women with a positive pregnancy test or those nursing infants were also excluded, as were diabetic patients taking any herbal or vitamin supplement were excluded. Similarly, subjects taking vitamin or herbal supplements were excluded to avoid the influence of these on the outcome of the results. All subjects came to the clinic between 8 A.M. and 10 A.M. This medical center is a charity hospital run by the State, and our population generally has a low socioeconomic status and low literacy background. The obesity incidence here is high in both the AA and white populations.

Blood was collected from study subjects after an overnight fast (8 h). EDTA tubes were centrifuged to collect clear plasma, as described previously [17]. Blood EDTA tubes were sent to the hematology laboratory for HbA1c analyses, and blood serum tubes were sent to the clinical chemistry lab of LSU Health Sciences Center for chemistry profile analyses. All plasma samples were stored at −80 °C. Plasma 25(OH)-vitamin D concentrations were determined using the sandwich ELISA method with kits purchased from Thermo Fisher Scientific Co. (Rockford, IL, USA). The company website has stated that the results of the 25-hydroxy-vitamin D assay using this ELISA kit are similar to the results obtained by mass-spectrometry chromatography. All analyses were carried out in duplicate. Results are expressed as µmole/mL plasma. The variability in duplicate assays was less than 7%. All controls and standards were used as specified by the manufacturer kit. Bone mineral density (g/cm^2^) was measured at the lumbar spine (L1–L4) using dual-energy X-ray absorptiometry (DXA) and a LUNAR DPX densitometer. Statistical analyses between the data in various groups were carried out using the Student’s T-test when equal variation was present or Welch’s T-test when equal variance was not assumed with the Sigma Plot statistical software 14.5 (Inpixon, Palo Alto, CA, USA). A two-tailed *p*-value less than 0.05 was considered significant.

## 3. Results

Table 1 provides subject demographics, including gender, age, BMI, duration of diabetes, and levels of HbA1c, TG, [25(OH)VD], [25(OH)VD]/TG, BMD, and calcium in the blood. The age, BMI, duration of diabetes, HbA1c, and calcium levels were similar between the AA–diabetic subjects and Whites–diabetic subjects groups. The number of white participants enrolled in this study was smaller relative to the African Americans because of the similar ethnic demographics of the city population. The serum levels of TG and 25(OH)VD were significantly lower in AA compared to whites. Since hypertriglyceridemia can be classified as mild (TG 150–199 mg/dL), high (TG 200–499 mg/dL), or very high (TG > 500 mg/dL), the TG levels for AA in our study were within the normal range while those for whites were on average in the high TG range [18,19], thus demonstrating the presence of the TG paradox in AA. The vitamin D/BMD paradox was also observed, as BMD levels were significantly higher in the AA group despite diminished levels of 25(OH)VD in these subjects. The 25(OH)VD levels, when expressed as a ratio of TG, were higher in AA, but the differences were not statistically significant (*p* = 0.12). Further analysis of the data after separating patients based on gender showed that AA females and white females showed a similar trend in elevated TG, 25(OH)VD, and 25(OH)VD/TG ratio levels. However, the number of male subjects in each group was limited and is, therefore, not discussed here.

## 4. Discussion

In this cross-sectional study of Type II diabetics, we found that AA had a significantly higher BMD, despite a more diminished vitamin D status, and significantly lower triglyceride levels compared to the white group, meaning the two paradoxical relationships commonly seen in AA were simultaneously present in this group of diabetic subjects. This study provides further evidence of these two paradoxical phenomena in AA while also providing possible explanations for them. In addition, our study is the first to show a positive correlation between vitamin D and BMD in AA and white subjects when vitamin D status is quantified as a ratio of serum [25(OH)D] to total serum TG. We believe this relationship may be due to a possible connection between these two paradoxes.

### 4.1. Rationale behind the Use of 25(OH)VD to Total Serum TG Ratio

Vitamin D is a lipophilic vitamin similar to vitamin E (VE). Clinical symptoms of VE deficiency were originally obtained by assessing plasma or serum concentration of VE as per ml serum [20,21,22,23,24,25,26,27,28,29]. However, it was concluded that this was not an accurate assessment of VE status after multiple clinical studies found that the ratio of serum VE to total serum lipids more accurately determined VE status and better predicted VE deficient symptomology in patients [30,31,32,33,34,35]. This finding led to the ratio of VE to lipid being used to clinically determine VE status [27,28,29,30,31,32,36,37,38] and also encouraged us to assess whether or not 25(OH)VD to total TG ratio is a better method in assessing vitamin D deficiency. Currently, vitamin D status is determined by quantifying circulating levels of 25(OH) vitamin D (25(OH)VD) per ml of serum or plasma [39,40,41,42]. To the authors’ best knowledge, no previous study has examined if the ratio of serum 25(OH)VD to serum lipids is more efficient in determining vitamin D status compared to the current method and if the inverse relationship among vitamin D deficiency and high BMD in AA retires when measuring vitamin D levels as this ratio described. The present study showed a positive correlation between vitamin D and BMD in AA and white subjects when vitamin D status was quantified as a ratio of serum [25(OH)D] to total serum TG. More studies are needed to determine whether or not the vitamin D to total TG ratio could better correspond with vitamin D deficient symptomology.

### 4.2. Brief Overview of Vitamin D Metabolism and Function

Vitamin D is a fat-soluble vitamin that is obtained through diet and sun exposure [43,44]. Vitamin D is difficult to obtain through diet due to the scarcity of foods that contain vitamin D. As a result, humans get most of their vitamin D, approximately 80%, through sun exposure [44]. Initially, vitamin D is synthesized as an inert precursor that undergoes two hydroxylation steps to be converted to its biologically active form. The first hydroxylation step occurs in the liver, where the inert form is converted to [25(OH)D], the major circulating form of vitamin D, which was used to assess vitamin D status in the present study. The second hydroxylation step occurs in the kidneys, where the molecule is converted into its active form, 1,25-dihydroxyvitamin D, commonly known as calcitriol [43].

Vitamin D has several important roles in the body, including signaling processes, expression and genetic responses, hormone protein synthesis, immune/inflammatory responses, and turnover and cell synthesis, including the most studied role in bone health [45]. Vitamin D acts in bone health via calcium and phosphorus metabolism [46] (Figure 1). Together, calcium and phosphorus constitute hydroxyapatite, an important component of bone [47]. More than 99% of the calcium in the body is found in bones and teeth, demonstrating its importance in maintaining bone health [48]. Approximately 85% of phosphorus is present in bone or teeth as hydroxyapatite, with the remaining 15% found in cells [49]. In the presence of vitamin D, calcium is absorbed in the small intestine, predominantly the ileum [50], and then distributed to the bones. When sufficient vitamin D levels are present, roughly 30–40% of the calcium entering the body is absorbed. This value drops to roughly 10–15% when inadequate levels of vitamin D are present, leading to decreased calcium levels in the bone [46,51]. Phosphorus, like calcium, also relies on vitamin D for absorption from the GI tract. As a result, deficient vitamin D levels can lead to hypophosphatemia due to decreased phosphorus absorption [49]. Parathyroid hormone (PTH), a peptide hormone secreted by the parathyroid glands, also regulates calcium and phosphate levels by its direct impact on vitamin D levels. PTH stimulates the synthesis of calcitriol, the active form of vitamin D, in the kidneys, leading to increased absorption of calcium and phosphorus (Figure 1). PTH levels are regulated through a negative feedback loop with calcium levels; therefore, when calcium levels increase, PTH secretion is inhibited. Vitamin D also exerts negative feedback on PTH [52].

### 4.3. Possible Explanations for the Vitamin D and BMD Paradox in AA

While the reasons for the vitamin D/BMD paradox in AA are still under debate, several hypotheses exist.

#### 4.3.1. Genetic Polymorphism and Vitamin D Binding Protein

Powe et al. concluded that although AA have lower levels of total vitamin D compared to whites, it is likely that the two races still have similar concentrations of bioavailable vitamin D due to a genetic polymorphism that results in decreased levels of vitamin D binding protein (VDBP) in AA [53]. Their study assessed total and bioavailable levels of vitamin D, VDBP and PTH levels, and BMD in AA (*n* = 1181) and whites (*n* = 904). Results showed that both total vitamin D and VDBP levels were lower in AA than in whites, levels of bioavailable vitamin D were similar between the two groups, and BMD was higher in AA than in whites. The authors found that 79.4% of the difference in VDBP levels between the two groups could be explained by a genetic polymorphism in the coding region of the VDBP gene at rs7041 and rs4588. In another study, it was also concluded that the measurement of bioavailable vitamin D is likely to be a more accurate way to determine vitamin D activity than the measurement of total vitamin D levels [54]. These two studies suggest that total vitamin D levels are not a reliable marker for detecting vitamin D deficiency. This may explain why AA did not show signs of vitamin D deficiency, such as low BMD, despite expressing decreased levels of total vitamin D [55,56,57]. However, a study carried out in 164 AA and white subjects concluded that there was no advantage to using free vs. total vitamin D levels to analyze vitamin D sufficiency and that no significant difference in VDBP levels existed between the two ethnic groups [6]. Due to the scarcity of studies and varying results, more convincing research is needed in this area before a strong supporting argument can be made for this hypothesis.

#### 4.3.2. Optimal Vitamin D Levels

Wright et al. suggested that adequate vitamin D levels may be lower for AA compared to whites [58]. The authors evaluated the relationship between vitamin D levels, measured as serum [25(OH)D], and intact PTH in AA (*n* = 423) and whites (*n* = 1258). As stated in the Introduction, vitamin D inhibits PTH secretion via a negative feedback loop. This leads to a natural inverse relationship between vitamin D and PTH levels; for AA, the inverse relationship only existed at serum vitamin D levels < 23 ng/mL, but for whites, this relationship was present until vitamin D levels reached 32 ng/mL, demonstrating that AA have a lower intact PTH threshold than whites. While opinions vary concerning what the optimal vitamin D level should be, serum vitamin D levels of 30 ng/mL are normally considered sufficient for bone health [59]. According to the study conducted by Wright et al., [59] this value is much closer to the PTH threshold level of whites than that of AA. These data suggest that optimal vitamin D levels may be different between AA and whites, an assertion also made by Cauley et al. [60].

#### 4.3.3. Skeletal Resistance to PTH

As stated previously, vitamin D exerts negative feedback on PTH secretion. Since vitamin D levels are consistently lower in AA, this results in AA having higher PTH concentrations than whites [61]. Since PTH induces bone resorption by osteoclasts, theoretically, AA should have a higher rate of bone breakdown and lower BMD than whites due to their greater PTH levels [61]. However, studies have shown the opposite; AA seem to have lower rates of bone resorption and increased rates of bone formation than whites [61,62,63]. A possible explanation for this finding is that AA have decreased sensitivity to PTH-induced bone resorption. In one study, 15 AA and 18 white pre-menopausal women were given continuous PTH infusions for 24 h, after which markers of bone resorption were analyzed. Results showed that whites had significantly higher markers of bone resorption than AA, leading the authors to conclude that when compared to whites, AA have decreased sensitivity to PTH [62]. Age difference has also been shown to influence PTH and vitamin D levels; however, the age range of participants in the present study is likely not large enough to affect our results [64].

#### 4.3.4. Calcium Economy

Calcium plays an important role in bone health and there is evidence that increased calcium absorption in AA during childhood and adolescence is the potential reason for the disconnect between BMD and vitamin D levels. Higher levels of PTH in AA, which is potentially an evolutionary phenomenon that resulted from adapting to life in the hot and dry climate of Africa, leads to increased renal absorption of calcium, resulting in decreased calcium excretion in the urine [2,56,65,66]. This increased calcium conversion explains why, despite lower calcium intakes, AA can maintain their muscle mass during maturity [65]. Bryant et al. reported a 57% higher rate of calcium retention in adolescent AA girls when compared to white girls of the same age [67]. Another study concluded that lower urinary calcium, due to increased calcium absorption, is the reason why AA have a more positive calcium balance and accounts for the racial difference in BMD [68]. One study assessing racial differences in calcium metabolism in 89 AA (*n* = 38) and white (*n* = 51) girls aged 4.9–16.7 years reported that the AA girls had significantly lower calcium urinary excretion and a greater bone calcium deposition than white girls [69]. It is worth noting that in our study, differences in calcium levels between whites and AA were not significant; despite decreased levels of vitamin D, AA still had adequate calcium levels. Together, these findings suggest that increased calcium absorption rates during childhood and adolescence may account for the greater bone mass seen in AA.

### 4.4. Brief Overview of Triglyceride Metabolism and Function

Triglycerides, a type of lipid found in adipose tissue, consist of three fatty acids bound to a single glycerol molecule (Figure 2). The primary function of triglycerides is energy storage; others include insulation, cushioning, and absorption of some vitamins [70]. After ingestion, triglycerides are hydrolyzed in the gastrointestinal tract, forming free fatty acids and 2-monoacylglycerols. These components then enter enterocytes, where they are reassembled into triglycerides and packaged into chylomicrons, one of the major types of lipoproteins [71]. Chylomicrons then exit the enterocyte at the basolateral surface and enter the lymphatic system downstream. After entering the systemic circulation and having apolipoprotein (APO) CII attached, the chylomicron then activates lipoprotein lipase (LPL), an enzyme that degrades triglycerides, resulting in the release of fatty acids that can be used for energy or stored [72]. When there is decreased LPL activity, the result is hypertriglyceridemia, a condition associated with adverse health outcomes [19].

### 4.5. Possible Explanations for the Triglyceride Paradox in AA

Possible reasons for the TG paradox in AA focus mainly on factors that affect LPL activity. Studies have shown AA to have increased LPL activity and to display similar levels of visceral adipose tissue despite having a higher fat content compared to whites. This results in AA having a greater concentration of high-density lipoprotein (HDL) and a more cardioprotective lipoprotein lipid profile than whites [73,74]. Decreased levels of apolipoprotein CIII, which inhibits LPL, have also been reported in AA [75]. Additionally, insulin resistance seems to inhibit LPL activity in whites but not in AA [76]. In essence, increased LPL activity in AA may explain the presence of the TG paradox (Figure 3). Whether or not differences in the dietary pattern between AA and whites, such as meat consumption, have a role in the TG paradox is not known and needs to be explored [66,77].

### 4.6. Our First Novel Hypothesis: Could the Paradoxical Relationships Be Linked to Each Other?

Currently, vitamin D status is determined by assessing circulating levels of [25(OH)D] per milliliter of serum or plasma [39,40,41,42]. To the best of our knowledge, no studies have examined whether measuring the ratio of serum vitamin D to serum lipids is more efficient in determining vitamin D status than the current method and whether the inverse relationship between vitamin D deficiency and high BMD in AA disappears when measuring vitamin D levels as this ratio described. We observed that AA had a higher BMD compared to whites but lower vitamin D levels when vitamin D status was quantified as [25(OH)D] per milliliter of serum. However, AA did not show vitamin D deficiency when vitamin D was expressed per TG. In addition, BMD levels correlated positively with vitamin D when the vitamin D status was quantified as a ratio of TG (r = 0.29; *p* = 0.026). Therefore, we hypothesize that vitamin D status is more accurately obtained when quantified as a ratio of serum vitamin D to total serum TG, as we observed a positive correlation between vitamin D and BMD when vitamin D is quantified in this manner. We believe our results may be explained by the TG paradox seen in AA, which we believe is potentially linked to the vitamin D/BMD paradox. A significant correlation between TG and VDBP has been shown to exist [78,79]. Additionally, VDBP has been shown to correlate with levels of vitamin E (VE), a fat-soluble vitamin that uses lipoproteins for transportation [79]. Multiple clinical studies have also found that the ratio of serum VE to total serum lipids more accurately determines VE status and better predicts VE deficient symptomology [30,31,32,33,34,35]. In a study of 2085 subjects, Powe et al. found that AA have decreased levels of total vitamin D and VDBP, resulting in adequate levels of bioavailable vitamin D, leading us to conclude that the most likely reason for the vitamin D paradox in AA was inappropriate assessment of vitamin D status [53]. These findings further support our novel hypothesis that the reason for the adequate BMD status in AA is potentially due to a connection between the TG and vitamin D/BMD paradoxes (Figure 4). In summary, the rationale for our hypothesis is that as TG levels decrease due to the TG paradox, it is likely that VDBP decreases as well, given their association. Decreased levels of VDBP result in normal levels of bioavailable vitamin D, resulting in an adequate bioavailable vitamin D status for AA despite their total vitamin D level being low. This differs from the traditional assessment of vitamin D status, which is determined by assessing circulating levels of [25(OH)D] per milliliter of serum or plasma because the current method ignores the role of VDBP, a molecule whose levels have been shown to be influenced by TG levels as stated prior and shown in Figure 4. Using this ratio that we describe to assess vitamin D status factors in the role VDBP exerts in determining bioavailable vitamin D levels, thus explaining why this ratio is a potentially more accurate measurement than the current method used.

### 4.7. Our Second Novel Hypothesis: Could Racial Differences in Estrogen Levels Be Responsible for the Vitamin D/BMD Paradox?

Estrogen is a hormone found in both males and females, with levels being higher in females. Estrogen serves several important functions, including reproductive and bone health. Estrogen exerts its effects on bone by activating osteoblasts, cells that produce new bone by secreting collagen, while inhibiting osteoclasts, cells derived from the macrophage lineage that break down bone. With menopause, estrogen levels decline in females, leading to decreased bone mineral density via decreased osteoblast activity and increased osteoclast activity, thus increasing one’s risk for osteoporosis [65,66].

Multiple studies have shown that AA have increased estrogen levels compared to other races [67,68,69,70]; however, no studies have suggested this as a possible reason for the vitamin D/BMD paradox. Another relationship worth mentioning is that between vitamin D and estrogen. From a biochemical perspective, vitamin D and estrogen both belong to the steroid family. Androgens are converted to estrogen by the aromatase enzyme (Figure 5). Increased vitamin D levels have been linked to the downregulation of androgen synthesis, leading to decreased estrogen levels [71,72]. Vitamin D has also been shown to inhibit estrogen receptor alpha, leading to an attenuation of estrogen signaling [72]. Therefore, one can expect that if vitamin D levels are decreased, this might lead to increased estrogen synthesis, which could possibly explain the increased BMD status in vitamin D-deficient individuals since estrogen exhibits an anabolic effect on bone (Figure 5). However, results are conflicting, since other studies have reported either no relation or a positive correlation between vitamin D and androgen or estrogen levels [73,74,75]. Androgen and estrogen levels between the two groups in the present study were also not assessed. As a result, further studies need to investigate this idea to see if increased estrogen levels in AA may be responsible for the vitamin D/BMD paradox in this population.

### 4.8. Our Third Novel Hypothesis: Could Racial Differences in Hydrogen Sulfide Levels Be a Potential Factor in the Vitamin D/BMD Paradox?

Hydrogen sulfide (H_2_S) is a gasotransmitter that has been shown to play a central role in various physiological processes and diseased states, including diabetes [80]. Two main enzymes, cystathionine β-synthase (CBS) and cystathionine-γ-lyase (CSE), govern the production of H_2_S. We have previously shown H_2_S levels to be decreased in AA [81]; other studies have also reported similar findings [82,83]. We have also shown vitamin D supplementation to increase H_2_S status by upregulating CSE in monocytes, thus making us believe that decreased H_2_S levels in AA may potentially be due to their diminished vitamin D status since vitamin D and H_2_S levels are shown to parallel each other [81]. Adiponectin is a hormone secreted by adipocytes that has been shown to stimulate osteoclast, resulting in bone resorption and decreased BMD status [84,85,86]. We have previously reported a positive correlation between adiponectin and H_2_S [87]. Together, this information leads us to hypothesize that AA, having naturally decreased vitamin D and H_2_S levels, leads to decreased adiponectin, resulting in increased BMD via decreased osteoclast activity (Figure 6). Hao et al. recently reported an inverse association between H_2_S levels and bone health [88], thus providing further support for this hypothesis. However, reports of opposite measures have also been reported [89,90,91]; as a result, more research needs to be conducted on this topic to see if it potentially explains the vitamin D/BMD paradoxical relationship.

### 4.9. Our Fourth Novel Hypothesis: HMG-CoA Reductase as a Possible Contributor to the Triglyceride Paradox

HMG-CoA reductase, primarily known for its role as the rate-limiting enzyme for cholesterol biosynthesis, potentially houses a connection with the TG paradox that has not been mentioned before. In 2009, Chen et al. conducted a prospective multi-ethnic cohort study consisting of 2444 subjects. From their data, they concluded that AA had a certain HMG-CoA reductase gene variant that resulted in decreased TG levels compared to other races [92]. To the authors’ best knowledge, nobody has mentioned this as a possible reason for the TG paradox. More studies should evaluate this idea and see if there is a possible connection between this enzyme and the TG paradox.

### 4.10. Why Is Studying These Paradoxical Relationships Important?

While these paradoxical relationships may not seem important, ignoring them may result in financial and health consequences for AA. Vitamin D supplementation among the general population has surged over the last several decades [93,94]. With the paradoxical relationship that exists between vitamin D levels and BMD, it is likely that AA showing decreased vitamin D levels on a serum screening test may not need to be advised to supplement with vitamin D to decrease their chance of fractures. Advising vitamin D supplementation to AA who express a below-normal vitamin D status on serum screening, based solely on the idea that it will enhance their bone health, may cause them to waste money on a supplement they do not need. However, it should be mentioned that vitamin D supplementation for all races can provide many other health benefits besides a potential increase in bone health [2], meaning just because it might not be beneficial for improving bone health in AA does not mean it should not be given.

Concerning the TG paradox, ignoring this may lead to the underdiagnosis of metabolic syndrome in AA [10]. Although the exact definition and criteria for metabolic syndrome have changed over the last 25 years [95], serum TG levels ≥ 150 mg/dL is one criterion still used to diagnose metabolic syndrome [95,96,97,98,99,100]. Since AA experience this TG paradox, it is likely that TG levels ≥ 150 mg/dL are not a reliable marker to use when trying to diagnose them with metabolic syndrome, especially when studies have shown that insulin resistance is present in AA with TG levels < 150 mg/dL [101,102,103,104]. Based on these data, it is possible that the TG diagnostic criteria used to diagnose metabolic syndrome in AA needs to be changed. This may also explain the paradoxical relationship that exists between the prevalence of metabolic syndrome and the occurrence of cardiovascular disease and Type II diabetes in AA; compared to whites, AA have a higher incidence of cardiovascular disease and Type II diabetes despite having a lower prevalence of metabolic syndrome [10,101,105,106,107]. If future studies conclude that lowering the TG cutoff value for diagnosing metabolic syndrome in AA is more accurate, this would likely cause a direct increase in the number of AA with metabolic syndrome. This would then abolish the unexplained inverse relationship currently seen when looking at the prevalence of metabolic syndrome and the occurrence of cardiovascular disease and Type II diabetes in AA.

## 5. Limitations of the Study

This manuscript discusses valuable and novel literature in the realm of an important topic, but limitations do exist. One limitation of this study is the limited number of subjects in each group. Another limitation is that the dietary patterns of the subjects in our study were not assessed. Differences in dietary habits of AA and whites are important risk factors for diabetes. Certain dietary behaviors, such as a high intake of red meat, have been associated with Type II diabetes and increased TG levels [108,109]. Since dietary patterns were not assessed, we are unable to say that the variance in the TG levels between the two groups was not due to a difference in their diet. If further investigation of these subjects revealed that whites ate more red meat and fatty foods than the AA group, then this may explain why these white subjects had higher TG levels; however, AA on average eat the most meat compared to other races, including whites, Hispanics, and East Asians [110] and have more unhealthy eating habits compared to whites [111,112,113,114], so if we did assess dietary patterns in these subjects, it is more probable that the AA group would have a higher TG intake compared to the white subjects, though this cannot be said for certain. Another limitation is the BMI of the subjects in our study. For both groups, the average BMI was in the obese range (>30), which is not representative of the general population. Lastly, while our novel hypotheses on H_2_S, estrogen, and HMG-CoA reductase as having potential roles in these paradoxes are intriguing, none of these factors were assessed in our patients to see if differences exist between the two groups. To control for the variables mentioned, future studies should consist of a greater quantity of patients with an equal amount of white and AA subjects and males and females but with various BMI levels. In addition, the dietary patterns for all subjects must be similar enough to where any difference in the results cannot be explained by a variance in diet. Each AA subject should also be matched with a white subject that shares a similar BMI with them, as this would allow us to see if BMI plays a role in these paradoxical relationships, and having different BMI levels, and not just BMI levels in the obese category, would be more representative of the general population. Studies assessing the levels of H_2_S and estrogen and the variant of HMG-CoA reductase in the subjects should also be considered to see if these variables may be responsible for these two paradoxical relationships, as mentioned previously.

## 6. Conclusions

In this study, we compared blood markers in AA and white subjects with Type II diabetes and found paradoxical relationships to exist between vitamin D and BMD status and TG levels in AA, thus providing increased evidence for the existence of these paradoxical relationships. To the best of our knowledge, this is the first study to simultaneously examine these relationships in a single group of Type II diabetic subjects. In addition, a review of the current hypotheses concerning why and whether these relationships exist was conducted, and novel hypotheses of our own were added. We also explained how ignoring one of these relationships might result in the underdiagnosis of metabolic syndrome in AA. As a result, we advocate for changes to the diagnostic criteria of TG status for metabolic syndrome in AA and explain that this can potentially provide a more accurate clinical representation of metabolic syndrome in AA and help us understand the current paradoxical relationship that exists between the prevalence of metabolic syndrome and the occurrence of cardiovascular disease and Type II diabetes in AA. This study shows that many unanswered questions remain with respect to these paradoxical relationships; further research is required to determine why they exist, as understanding this can potentially lead to advancements in clinical practice by allowing physicians to understand how vitamin D levels may differ among different races which can enhance the accuracy of clinical assessment and judgment of vitamin D status by considering people’s race. Future studies also need to assess whether using the vitamin D to total TG ratio better relates to symptoms of vitamin D deficiency compared to the current method used to determine vitamin D status.

## Figures and Tables

**Figure 1 ijms-25-01305-f001:**
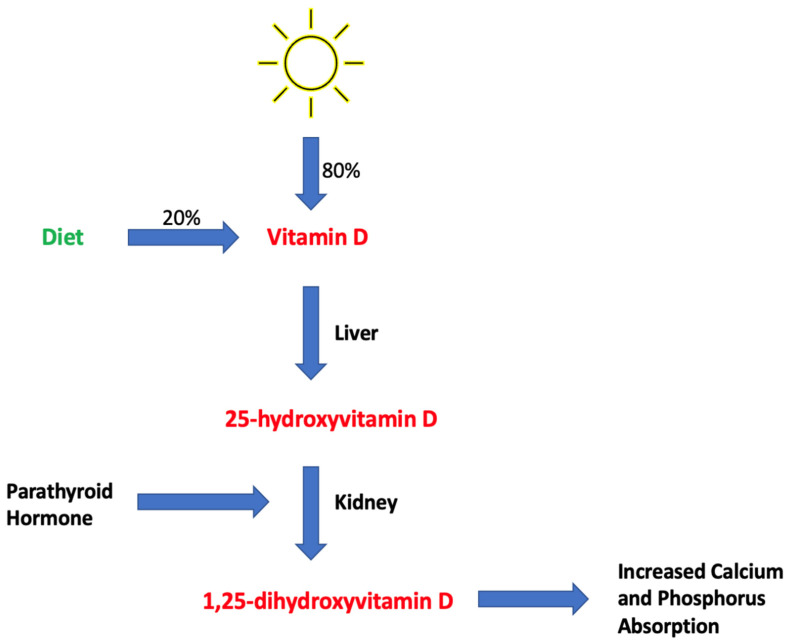
The biological process by which vitamin D is hydroxylated twice and converted into 1,25-dihydroxyvitamin D, resulting in increased calcium and phosphorus absorption.

**Figure 2 ijms-25-01305-f002:**
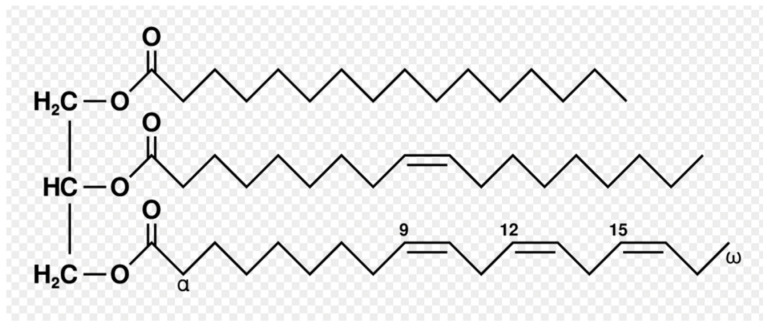
Structure of a single triglyceride molecule, consisting of three fatty acids with a glycerol unit as the backbone. https://upload.wikimedia.org/wikipedia/commons/b/be/Fat_triglyceride_shorthand_formula.PNG (accessed on 14 July 2023).

**Figure 3 ijms-25-01305-f003:**
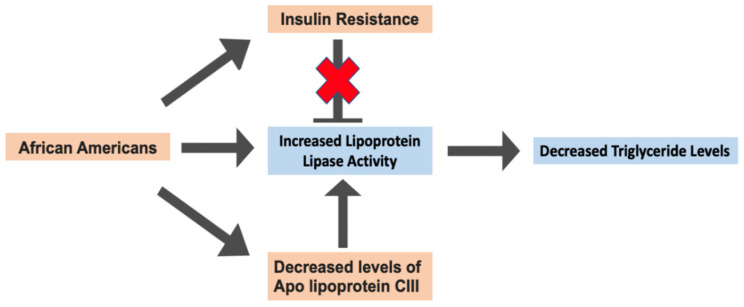
Diagram summarizing the potential reason for the triglyceride paradox in AA.

**Figure 4 ijms-25-01305-f004:**
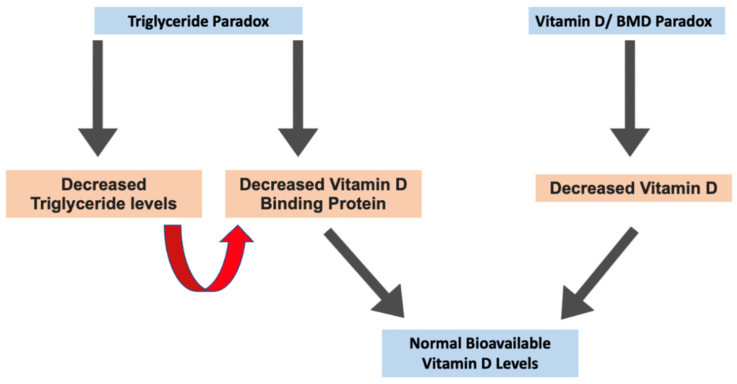
Summary showing the possible connection between the TG and vitamin D/BMD paradoxes.

**Figure 5 ijms-25-01305-f005:**
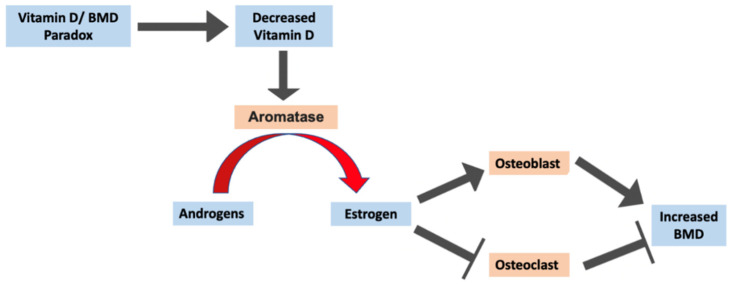
Summary diagram expressing the possible pathway in which increased estrogen levels in AA could be responsible for the vitamin D/BMD paradox.

**Figure 6 ijms-25-01305-f006:**
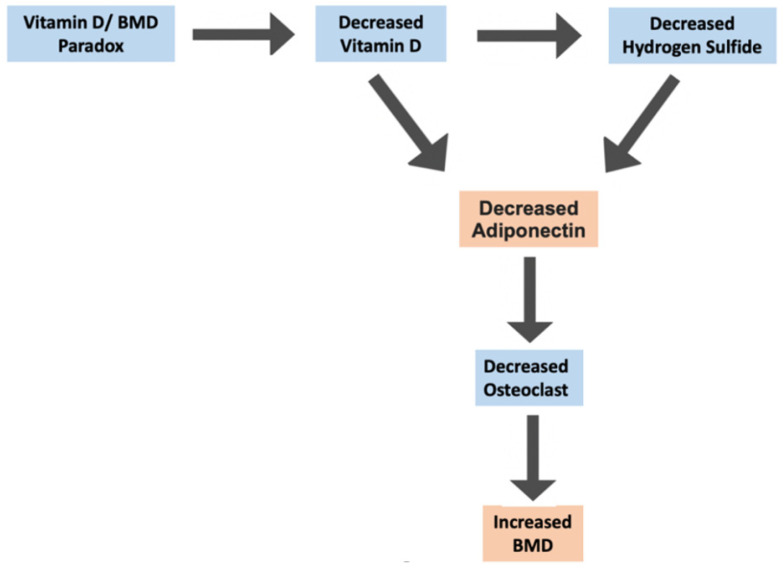
Summary diagram expressing the possible pathway in which decreased hydrogen sulfide levels in AA could be responsible for the vitamin D/BMD paradox.

**Table 1 ijms-25-01305-t001:** Various blood biomarkers in African American (AA) and white Type 2 diabetic subjects. Differences (mean ± SE) between values * vs. **, # vs. ##, and ^ vs. ^^ are significant (*p* < 0.05).

	AA	White	AA	White	AA	White
	Male + Female		Female		Male	
n	56	26	43	20	13	6
Age (yrs)	48 ± 1.3	53 ± 1.5	47 ± 1.5	52 ± 1.9	48 ± 3.2^	55 ± 0.8 ^^
BMI (kg/m^2^)	37 ± 1.2	38 ± 2.1	38 ± 1.3	37 ± 2.2	31 ± 1.8	40 ± 6.1
Diabetes duration (yrs)	3.7 ± 0.4	4.7 ± 0.9	3.6 ± 0.5	4.2 ± 1.0	3.7 ± 1.0	6.4 ± 2.4
HbA1c (%)	7.9 ± 0.3	7.4 ± 0.3	7.8 ± 0.3	7.0 ± 0.2	8.3 ± 0.5	8.5 ± 0.8
Triglycerides(mg/dL)	135 ± 15 *	243 ± 32 **	117 ± 7 #	242 ± 40 ##	196 ± 59	245 ± 53
25(OH)D (μM/mL)	15 ± 1.1 *	19 ± 1.9 **	15 ± 1.4 #	20 ± 2.2 ##	15 ± 1.5	15 ± 3.6
25(OH)D/Triglycerides	14 ± 1.8	8.6 ± 1.0	13 ± 2.0	9.0 ± 1.2	16 ± 4.0	6.9 ± 1.4
L1-L4 BMD (g/cm^2^)	1.3 ± 0.0 *	1.2 ± 0.0 **	1.3 ± 0.0 #	1.2 ± 0.0 ##	1.4 ± 0.0 ^	1.2 ± 0.0 ^^
Calcium (mg/dL)	9.4 ± 0.1	9.2 ± 0.1	9.3 ± 0.0	9.3 ± 0.1	9.6 ± 0.1 ^	9.2 ± 0.1 ^^

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
