# Peer review of "Vitamin D/Bone Mineral Density and Triglyceride Paradoxes Seen in African Americans: A Cross-Sectional Study and Review of the Literature"

_ijms, 2024, doi:10.3390/ijms25021305_

Round 1
Reviewer 1 Report
Comments and Suggestions for Authors
The article presents an interesting exploration of the relationship between vitamin D, bone mineral density (BMD) and triglyceride levels in African Americans (AA) with Type II diabetes. Overall, the abstract is well-constructed and informative. However, there are a few points and suggestions that could be considered for improvement:
1. ABSTRACT:
1. The abstract does not clearly state the methodology or design of the study. Providing information on the study design and sample size would enhance understanding of the research.
2. The abstract mentions the presence of the two paradoxical relationships in a single group of AA Type II diabetic subjects but does not provide specific findings or results. Adding a brief summary of key results would improve the abstract's completeness.
3. The term "TG paradox" may not be universally recognized and should be define prior to use as abbreviation
2. INTRODUCTION:
1. The introduction lacks some relevant and recent references about epidemiologic. Authors are suggested to add specific citations to enhance the quality and validity of the study.
2. The quantification of vitamin D is mentioned as "measured as serum 25-hydroxyvitamin D [25(OH)D] uM/ml." While this provides information on the method, it might be beneficial to include a brief explanation of the significance and its relevance to bone health.
3. Authors mentioned in introduction about "novel hypotheses" without providing a brief explanation of these hypotheses. Including a sentence or two about the specific content of these novel hypotheses would generate interest and anticipation.
4. The transition between discussing the paradoxes and introducing the study's objective could be smoother. Clearly outlining the research question and objectives immediately after introducing the paradoxes would enhance the logical flow of the introduction.
3. METHODS:
1. Authors mentioned that ethical approval was obtained but the specific details such as the name of the institution, the date of approval and the approval number are not provided. Authors must add ethical approval number for validity of the work. Authors should briefly mentioned about the consent from the patients/participants. Inclusion and exclusion criteria should be provided. The exclusion criteria are listed, but the rationale behind each criterion is not explained. For instance, why were patients with a history of cardiovascular disease, sickle cell disease, insulin treatment, or metabolic disorders excluded?
2. Authors describes the method for measuring plasma 25(OH)-vitamin D concentrations using ELISA but it does not specify the units of measurement (e.g., ng/mL) or the assay's sensitivity.
3. Authors described the data analysis using Sigma Plot software 14.5 but it lacks details on the statistical methods employed or any specific analyses conducted. Authors are suggested to briefly explain the statistical tests applied.
4. RESULTS:
Results are poorly described and should be reconsidered after robust statistical analyses.
5. DISCUSSION:
1. The use of the ratio of serum [25(OH)D] to total serum TG for quantifying vitamin D status is an interesting approach. However, it would be beneficial to discuss the rationale behind this choice more explicitly, explaining why this ratio is considered a more accurate measure than traditional methods.
2. The limitation section mentions that dietary patterns were not assessed and this could be a potential confounding factor. Authors should consider discussing the impact of dietary habits on the observed relationships and how future studies might consider controlling for this variable. Furthermore, explore whether BMI or other anthropometric measures were considered in the analysis.
3. In the conclusion, authors should discuss any potential implications for clinical practice.
4. Authors should provide future perspective of current study.
Author Response
Point by point reply to reviewers comments is given in the submitted rebuttal file.
We appreciate the time spent and many valuable suggestions given by the reviewers and the editor.

Reviewer 2 Report
Comments and Suggestions for Authors
In conclusion, Stevens and Jain's cross-sectional study contributes significantly to the understanding of paradoxical relationships in AA with Type II diabetes. The meticulous exploration of vitamin D, BMD, and TG status, coupled with novel hypotheses, positions this work as a valuable resource for researchers and clinicians alike. The study not only unravels existing paradoxes but also prompts a reevaluation of conventional assumptions, inviting a deeper exploration of these intricate physiological relationships.
The authors address a significant research gap by conducting a cross-sectional study comparing blood markers in individuals of African American (AA) descent and white individuals with type 2 diabetes. The methodology is thoroughly described, detailing participant registration, exclusion criteria, and blood sampling procedures.
The conclusion emphasizes the novelty of the study, being the first to simultaneously investigate these paradoxical relationships within one group of type 2 diabetes patients. The authors advocate for changes in the diagnostic criteria for metabolic syndrome in AA based on their findings, highlighting the potential impact on the risk of underdiagnosis. The article calls for further research to uncover the underlying causes of these paradoxes and their implications for healthcare development.
In summary, Stevens and Jain present a well-constructed and insightful article that provides valuable information to comprehend the paradoxical associations between AA individuals and type 2 diabetes. The study results suggest a reassessment of diagnostic criteria and underscore the need for additional research to unravel the mysteries surrounding these intriguing health differences.
Author Response
We again thank the reviewer for the time spent and for providing many valuable suggestions.

Round 2
Reviewer 1 Report
Comments and Suggestions for Authors
I am pleased to recommend the acceptance of the manuscript for publication as the authors have diligently addressed all the comments and concerns raised during the review process. The revisions have significantly improved the quality and clarity of the article.